# Alterations in Alzheimer’s Disease-Associated Gene Expression in Severe Obstructive Sleep Apnea Patients

**DOI:** 10.3390/jcm8091361

**Published:** 2019-09-01

**Authors:** Hsueh-Yu Li, Ming-Shao Tsai, Chung-Guei Huang, Robert Y. L. Wang, Li-Pang Chuang, Ning-Hung Chen, Chi-Hung Liu, Cheng-Ming Hsu, Wen-Nuan Cheng, Li-Ang Lee

**Affiliations:** 1Department of Otolaryngology—Head and Neck Surgery, Linkou Chang Gung Memorial Hospital, Taoyuan 333, Taiwan; 2Faculty of Medicine, College of Medicine, Chang Gung University, Taoyuan 333, Taiwan; 3Department of Otolaryngology—Head and Neck Surgery, Chiayi Chang Gung Memorial Hospital, Chiayi 613, Taiwan; 4Graduate Institute of Clinical Medical Sciences, College of Medicine, Chang Gung University, Taoyuan 333, Taiwan; 5Health Information and Epidemiology Laboratory, Chiayi, Chang Gung Memorial Hospital, Chiayi 613, Taiwan; 6Department of Laboratory Medicine, Linkou Chang Gung Memorial Hospital, Taoyuan 333, Taiwan; 7Research Center for Emerging Viral Infections, Chang Gung University, Taoyuan 333, Taiwan; 8Graduate Institute of Biomedical Sciences, Department of Medical Biotechnology and Laboratory Science, College of Medicine, Chang Gung University, Taoyuan 333, Taiwan; 9Department of Pulmonary and Critical Care Medicine, Taoyuan Chang Gung Memorial Hospital, Taoyuan 333, Taiwan; 10Stroke Center and Department of Neurology, Linkou Chang Gung Memorial Hospital, Taoyuan 333, Taiwan; 11School of Traditional Chinese Medicine, College of Medicine, Chang Gung University, Taoyuan 333, Taiwan; 12Department of Sports Sciences, University of Taipei, Taipei 111, Taiwan

**Keywords:** Alzheimer’s disease, inflammation, obstructive sleep apnea, transcriptome

## Abstract

Background: Obstructive sleep apnea (OSA) increases the risk of Alzheimer’s disease (AD), and inflammation may be involved in the early pathogenesis of AD in patients with OSA. However, the potential pathways between OSA and AD have yet to be established. In this study, we aimed to investigate differential expressions of AD-associated genes in OSA patients without evident AD or dementia. Methods: This prospective case-control study included five patients with severe OSA and five age and sex-matched patients with non-severe OSA without evident dementia who underwent uvulopalatopharyngoplasty between 1 January 2013 and 31 December 2015. The expressions of genes associated with AD were analyzed using whole-exome sequencing. Unsupervised two-dimensional hierarchical clustering was performed on these genes. Pearson’s correlation was used as the distance metric to simultaneously cluster subjects and genes. Results: The expressions of *CCL2*, *IL6*, *CXCL8*, *HLA-A*, and *IL1RN* in the patients with severe OSA were significantly different from those in the patients with non-severe OSA and contributed to changes in the immune response, cytokine–cytokine receptor interactions, and nucleotide-binding oligomerization domain-like receptor signaling pathways. Conclusions: Inflammation may contribute to the onset of AD and physicians need to be aware of the potential occurrence of AD in patients with severe OSA.

## 1. Introduction

Obstructive sleep apnea (OSA) can cause intermittent hypoxia, reoxygenation, hypercapnia, hypocapnia, changes in cerebral blood flow, and sleep fragmentation, which may result in brain disease [1]. The association between OSA and Alzheimer’s disease (AD) has been suggested based on population-based data and meta-analysis studies [2,3,4,5,6]. Furthermore, the potential pathways and causal relationships between OSA and AD have yet to be established [7].

In the last decade, “high-throughput” transcriptome investigation technologies have been developed to identify co-regulated gene networks linked with biological pathways and potentially modulating disease predisposition, outcomes, and progression [8]. For example, early-onset AD (EOAD), constituting barely 1–2% of all cases of AD and being diagnosed before the age of 65 years, is usually autosomal dominant inherited with genes including *APP* [9], *FKBP1B* [10], *PSEN1* [11,12] and *PSEN2* [11,12] being regarded as major factors. In comparison, the association between late-onset AD (LOAD) and genetic variants is much more complex, with at least 56 LOAD-associated genes [9,13,14,15,16,17,18,19,20,21,22,23,24,25,26,27,28,29,30,31,32,33,34,35,36] reported in the literature. These genes may involve lipid metabolism and regulate the production and clearance of amyloid β, thereby inducing AD [37]. Previous studies also showed an association of OSA and AD-related biomarkers, including brain amyloid β load and cerebrospinal fluid levels of tau [38,39,40,41,42].

In this study, we aimed to investigate differential expressions of genes associated with AD using whole-exome sequencing in middle-aged patients with OSA without evident AD using a case-control study.

## 2. Methods

### 2.1. Ethical Considerations

This study was approved by the Institutional Review Board of the Chang Gung Memorial Foundation (101-3547A3 and 102-5609A3), and written informed consent was obtained from all participants.

### 2.2. Participants

We prospectively recruited 10 consecutive otherwise healthy patients without memory deficits (five severe OSA cases with an apnea–hypopnea index [AHI] ≥30 events/h and five non-severe OSA controls with an AHI <15 events/h) who underwent uvulopalatopharyngoplasty in a tertiary care hospital between 1 January 2013 and 31 December 2015 [43]. None of the patients had a Mini-Mental State Examination score <25 [44], stroke, or heart failure after 2 years of follow-up. Resected submucosal tissues were obtained intraoperatively from the uvula.

### 2.3. RNA Isolation of the Study

Samples were immediately rinsed with phosphate-buffered saline to remove excess blood, minced, subjected to RNA extraction, and stored at −80 °C. We isolated and sequenced the total RNA of resected submucosal tissues of the uvula using RNA-Seq technology [45]. In brief, total RNA from cells of identical density was isolated using guanidinium phenol reagent (TRIzol reagent; Life Technologies, Grand Island, NY, USA) according to the manufacturer’s instructions. After synthesis, purification, and 3’-end single nucleotide A (adenine) addition of double-strand cDNA, the fragments were enriched by PCR amplification. During the quality control step, an Agilent 2100 Bioanaylzer (Agilent Technologies, Santa Clara, CA) and ABI StepOnePlusTM Real-Time PCR System (Applied Biosystems, Foster City, CA) were used to qualify and quantify the sample library. The library products were ready for sequencing via an Illumina HiSeqTM 2000. On average, 22,415,673 raw sequencing reads were generated and then 22,181,205 clean reads were obtained after filtering those of low quality. After obtaining raw data, quality control of alignment was performed to determine whether resequencing was needed.

### 2.4. Bioinformatics Analysis

After passing quality control, the alignment data were used to calculate the distribution of reads on reference genes and mapping ratio. We then performed downstream analysis including gene expression and deep analysis based on the gene expression. Gene quantification was performed using RNA-Seq by Expectation Maximization [46] to determine which transcripts were isoforms of the same gene, and gene expression levels were calculated using the method of fragments per kilobase of transcript per million map reads for direct comparisons of differences in gene expressions among samples. Gene expression values across all subjects were normalized as follows: Gnorm = (expression level of the gene for each subject – average expression level of the gene across all subjects)/standard deviation.

Differentially expressed genes (DEGs) were screened using the Poisson distribution method [47], and correction for type I and II errors was performed using the false discovery rate (FDR) method [48]. The Q value (error ratio) was calculated as the number of false positive genes/the number of selected DEGs. More stringent criteria (FDR ≤ 0.001 and an absolute value of Log2Ratio ≥ 1) were applied to identify DEGs. For our analysis, 63 genes (for details, see Appendix A) including four EOAD-associated genes (*APP* [9], *FKBP1B* [10], *PSEN1* [11,12], and *PSEN2* [11,12]) and 56 LOAD-associated genes (*ABCA1* [13,14], *ABCA7* [13,14], *ANK1* [15], *APOC1* [16], *APOE* [17], *BACE2* [18], *BCR* [16], *BIN1* [19], *CASS4* [20], *CCL2* (also known as monocyte chemoattractant protein 1 [MCP-1]) [21], *CD2AP* [13,22], *CD33* [13,22], *CELF1* [20], *CLU* [23,24], *COX4I1* [25], *CR1* [24], *CXCL8* [35], *DSG2* [20], *EPHA1* [13,22,26], *EPHA10* [13,22,26], *FAM136A* [16], *FERMT2* [20], *GAS7* [27], *GRIK2* [16], *GRN* [28], *HAS3* [16], *HLA-A* [36], *HLA-DRB1* [20], *HLA-DRB5* [20], *IL1RN* [34], *IL6* [29], *INPP5D* [20], *ITGAL* [16], *KLK2* [16], *LRP2* [30], *MAPT* [28], *MEF2C* [20], *MS4A4* [13,19,22], *MS4A6E* [13,19,22], *PICALM* [23], *PILRA* [16], *PIN1* [32], *PLD3* [9], *PRSS42* [16], *PRSS45* [16], *PTK2B* [20], *RIN3* [20], *SEC31A* [16], *SLC22A2* [16], *SLC24A4* [20], *SORL1* [20], *STOX2* [16], *THNSL2* [16], *TMIE* [16], *TRDMT1* [16], *TREM2* [33] and *ZCWPW1* [20]) were selected from well-established curated resources, including published literatures, the Database for Annotation, Visualization and Integrated Discovery (DAVID) (version 6.8; https://david.ncifcrf.gov/) [49], Gene Association Database (GAD) (https://geneticassociationdb.nih.gov/), and Kyoto Encyclopedia of Genes and Genomes (KEGG) (http://www.genome.jp/kegg/) [50]. We performed a deep analysis based on DEGs, including GAD, Gene Ontology (GO) enrichment analysis, and KEGG pathway enrichment analysis. 

### 2.5. Statistical Analysis

Unsupervised two-dimensional hierarchical clustering was performed on these 63 unique genes. Pearson’s correlation was used as the distance metric to simultaneously cluster subjects (based on their global expression profile) and genes (based on expression levels across subjects). The calculated *p*-values of bioinformatics analysis were subjected to Benjamini correction, taking an adjusted *p*-value < 0.05 as a threshold. All analyses were performed using SAS software version 9.4 (SAS Institute, Cary, NC, USA) and SPSS software version 24 (International Business Machines Corp., Armonk, NY, USA), and the level of statistical significance was set at *p* < 0.05.

## 3. Results

### 3.1. Participant Characteristics

The patient characteristics are shown in Table 1. The patients were relatively younger, mostly male and non-obese, but there were no significant differences between the two groups in age, male sex, body mass index, minimal SpO_2_, or time with SpO_2_ < 85%. As expected, the patients with severe OSA had a significantly higher mean AHI (mean, 60.6; SD, 21.2) compared to those with non-severe OSA (mean, 7.0; SD, 4.3) (effect size = 3.50). None of the participants had a diagnosis of the comorbidities associated with AD.

### 3.2. AD-Associated Transcriptomic Expression in the Uvular Tissue of the Patients with Severe OSA 

Unsupervised two-dimensional hierarchical clustering of the expression profiles from over 60 AD-associated genes did not differentiate the patients with severe OSA from those with non-severe OSA (Figure 1). We identified five AD-associated DEGs including one up-regulated gene (*IL1RN*) and four down-regulated genes (*CCL2*, *CXCL8*, *HLA-A*, and *IL6*) at a FDR of 0.001 (Table 2). None of the EOAD-associated genes were significantly differently expressed. Detailed comparisons of the expressions of the 60 AD-associated genes between the two groups in descending order by probability are shown in Appendix A. 

Highly enriched modules, relating to these five DEGs in the patients with severe OSA, included 27 biological processes, three cellular components, and three molecular functions (for details, see Appendix A). Furthermore, only six biological processes (immune response; cellular response to interleukin-1; cellular response to tumor necrosis factor; cellular response to lipopolysaccharide; regulation of vascular endothelial growth factor production; and negative regulation of IL-1-mediated signaling pathway) and one cellular component (extracellular space) reached statistical significance after Benjamini correction. 

We found a highly enriched AD module as derived from “GAD-disease” in the patients with severe OSA (five counts; DEGs with pathway annotation: 1.3%; unadjusted *p*-value = 4.3 × 10^−7^; Benjamini adjusted *p*-value = 4.6 × 10^−4^). However, functional annotation as derived from, the KEGG, did not reveal an enriched AD module (Table 3). Nevertheless, the nucleotide-binding oligomerization domain (NOD)-like receptor signaling pathway and cytokine–cytokine receptor interaction pathway were significantly enriched.

## 4. Discussion

In this study, five AD-associated DEGs were identified in our middle-aged patients with severe OSA without symptoms of AD. These findings suggest that preclinical changes in genetic expressions may contribute to the onset of AD in patients with severe OSA. The uvular tissue of the patients with severe OSA exhibited enriched biological processes that resulted in the change in state or activity of a cell in terms of movement, secretion, enzyme production, and gene expression because of an IL-1, tumor necrosis factor, or lipopolysaccharide stimulus. In the study, major genetic factors for EOAD (such as *APP*, *FKBP1B*, *PSEN1*, and *PSEN2*) and LOPD (such as *APOE*, *CLU*, *SORL1*, *PICALM*, and *BIN1* [9,10,11,12,51]) did not show significantly different expressions in the patients with severe OSA without evident AD. However, the expressions of *CCL2*, *IL6*, *CXCL8*, *HLA-A*, and *IL1RN* were significantly different in the patients with severe OSA compared to those with non-severe OSA. 

Previous studies [21,29,36,52] have reported significant associations between *CCL2*, *HLA-A2*, *IL6*, and *IL1RN* gene polymorphisms and *AD*, and *IL6* gene polymorphisms have been associated with the risk of adult OSA [53]. We previously found that the expression of the *CCL2* gene was significantly elevated in peripheral blood monocytes of patients with severe OSA [54], whereas its expression was significantly decreased in the uvular tissue in this study. Elevated serum levels of MCP-1, IL-1ß, and IL-6 in patients with OSA may also be risk factors for the development of AD [55,56,57]. The down-regulated expression of CXCL8 in patients with OSA has been reported to possibly reduce its protective effect against amyloid-ß-induced neurotoxicity [35]. Moreover, changes in the cytokine–cytokine receptor interaction pathway [58] or NOD-like receptor signaling pathway [59] may activate the NOD-like receptor 3 inflammasome and promote neuroinflammation and brain pathologies [59]. These observations suggest that inflammation links OSA and AD.

Although the prevalence of OSA in patients with dementia has been estimated to be high, little is known about the association between OSA and AD with regards to disease severity [60]. In a comparative study of nursing home patients, the severe OSA group had significantly lower dementia rating scales than the mild–moderate OSA group (31 vs. 127, *p* < 0.004) [61]. The severe OSA group (AHI ≥ 40 events/h) had a higher incidence of AD than those with less severe OSA (AHI < 40 events/h). Ancoli-Israel et al. also reported that the respiratory disturbance index was significantly correlated with total dementia rating scale score (r = −0.37, *p* < 0.001) [61]. In their study, the correlation could be influenced by different reliability in night-to-night respiratory disturbance index (r = 0.68) and dementia rating scale score (r = 0.97). After correction for attenuation, the correlation coefficient between respiratory disturbance index and dementia rating scale score was estimated to be approximately −0.46. These data suggest that severe OSA patients have a higher incidence of AD and lower cognitive function, and that there is a moderate association in disease severity between OSA and AD.

There are limitations to this study. First, although the use of molecular expression from excised uvular tissue may be a surrogate to correlate potential pathways of OSA-related AD, further research to compare the molecular expression in uvular tissue in AD patients is warranted to confirm our results. Second, this is a case-controlled study; therefore, we could not follow up these patients and investigate the occurrence of AD years later. Third, the sample size was small. However, this study proved the alterations in AD-associated gene expression in severe OSA patients. Based on our findings, inflammatory processes related to severe OSA may be possible mechanisms to prevent AD. Prospective clinical trials are necessary to elucidate the causal relationship between OSA and AD and, moreover, to clarify whether continuous positive airway pressure (CPAP) therapy for OSA affects the prevalence and onset of AD.

## 5. Conclusions

In our middle-aged patients with severe OSA without evident AD, the genetic alterations in the biological processes and pathways suggest that inflammation is a possible early change leading to the occurrence of AD. Our study extends the OSA disease spectrum and should serve to remind physicians to be aware of the potential occurrence of AD in patients with OSA. Further research is warranted to explore the genetic characteristics of OSA-AD and investigate the effect of CPAP therapy on the onset and occurrence of AD.

## Figures and Tables

**Figure 1 jcm-08-01361-f001:**
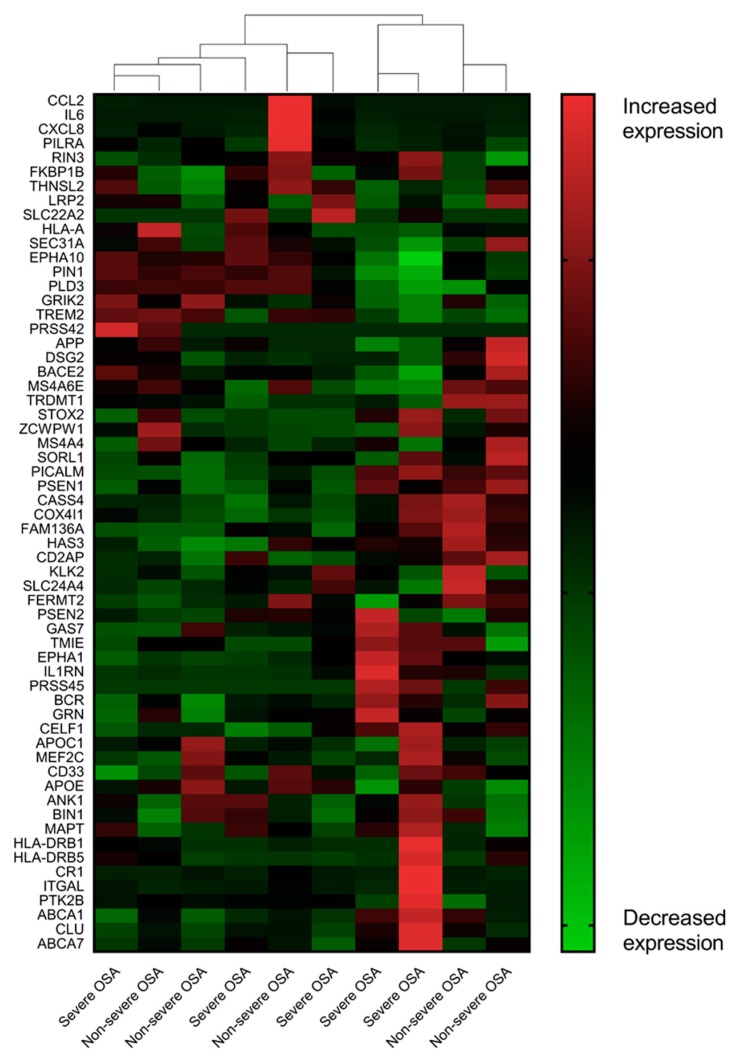
Alzheimer’s disease-associated transcriptome of the uvula tissue from the patients with severe OSA and non-severe OSA. In this analysis, we made no a priori designation of a subject’s phenotype (severe OSA, non-severe OSA control) when clustering individuals based on their oropharyngeal gene expression patterns. The results implied that severe sleep apnea elicited modest perturbations in the AD-associated transcriptional response of the oropharynx.

**Table 1 jcm-08-01361-t001:** Differences in patient characteristics between the severe obstructive sleep apnea (OSA) and non-severe OSA groups.

Variables	Severe OSA	Non-Severe OSA	Effect Size	*p*-Value
Patients	*n* = 5	*n* = 5		
Age (years), mean (SD)	34.6 (4.2)	41.2 (12.0)	0.73	0.07
Male sex, *n* (%)	5 (100)	4 (80)		>0.99
BMI (kg/m^2^), mean (SD)	26.4 (3.2)	25.5 (2.9)	0.30	0.07
AHI (events/h), mean (SD)	60.6 (21.2)	7.0 (4.3)	3.50	<0.001
Minimum SpO_2_ (%), mean (SD)	79.0 (6.0)	85.2 (2.3)	1.37	0.06
Time SpO2 < 85% (%), mean (SD)	1.66 (2.3)	0.04 (0.04)	1.00	0.15

Abbreviations: AHI: apnea–hypopnea index; BMI: body mass index; OSA, obstructive sleep apnea; SpO_2_: oxygen saturation measured by pulse oximetry.

**Table 2 jcm-08-01361-t002:** Expression of representative Alzheimer’s disease-associated genes among the patients with severe OSA and non-severe OSA in descending order by probability.

Gene Symbol	Expression		Log_2_(Fold Change)	Regulation	Probability
	Severe OSA	Non-Severe OSA			
Early-onset Alzheimer’s disease					
*APP*	382.418	587.644	−0.619	Down	0.503
*PSEN1*	12.776	14.876	−0.220	Down	0.286
*PSEN2*	7.204	6.208	0.215	Up	0.265
*FKBP1B*	3.786	4.126	−0.124	Down	0.216
Late-onset Alzheimer’s disease					
*CCL2*	12.992	116.716	−3.167	Down	0.868 ^a^
*IL6*	2.854	33.06	−3.534	Down	0.857 ^a^
*CXCL8*	2.042	20.058	−3.296	Down	0.824 ^a^
*HLA-A*	1.272	14.006	−3.460	Down	0.811 ^a^
*IL1RN*	239.35	57.236	2.064	Up	0.806 ^a^
*CLU*	490.412	308.192	0.670	Up	0.522
*BIN1*	23.218	18.884	0.298	Up	0.335
*APOE*	24.526	29.308	−0.257	Down	0.318
*SORL1*	8.136	9.338	−0.199	Down	0.267
*PICALM*	56.394	52.512	0.103	Up	0.233

Abbreviations: OSA, obstructive sleep apnea. ^a^ Significantly up- and down-regulated genes were defined as false discovery rate (FDR) ≤ 0.001 (i.e., probability ≥ 0.8) and an absolute value of Log_2_Ratio ≥ 1.

**Table 3 jcm-08-01361-t003:** Functional annotation chart of the Kyoto Encyclopedia of Genes and Genomes according to the five Alzheimer’s disease-associated differentially expressed genes in severe OSA patients.

Term	Count	Percentage	*p*-Value	Adjusted *p*-Value
Malaria	3	0.8	<0.001	0.007 ^a^
NOD-like receptor signaling pathway	3	0.8	<0.001	0.005 ^a^
Rheumatoid arthritis	3	0.8	<0.001	0.008 ^a^
Chagas disease	3	0.8	<0.001	0.008 ^a^
Influenza A	3	0.8	0.002	0.02 ^a^
Herpes simplex infection	3	0.8	0.002	0.02 ^a^
Cytokine–cytokine receptor interaction	3	0.8	0.003	0.02 ^a^
Graft-versus-host disease	2	0.5	0.01	0.09
Legionellosis	2	0.5	0.02	0.12
Pertussis	2	0.5	0.03	0.15
Salmonella infection	2	0.5	0.04	0.15
Amoebiasis	2	0.5	0.045	0.18
TNF signaling pathway	2	0.5	0.045	0.18
Toll-like receptor signaling pathway	2	0.5	0.045	0.18
Hepatitis B	2	0.5	0.06	0.22
Non-alcoholic fatty liver disease	2	0.5	0.06	0.21
Transcriptional misregulation in cancer	2	0.5	0.07	0.22
Chemokine signaling pathway	2	0.5	0.08	0.23

Abbreviations: NOD, nucleotide-binding oligomerization domain; OSA, obstructive sleep apnea; TNF, tumor necrosis factor; ^a^
*p*-values < 0.05 adjusted using the Benjamini method were statistically significant.

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
