# Peer review of "Alterations in Alzheimer’s Disease-Associated Gene Expression in Severe Obstructive Sleep Apnea Patients"

_jcm, 2019, doi:10.3390/jcm8091361_

Round 1

Reviewer 1 Report

The present paper examined expressions of genes associated with AD, analyzed using whole-exome sequencing, in five participants with severe OSA and five age and sex-matched patients with non-severe OSA, none of which had evident dementia. The results suggest preclinical changes in genetic expression in those with severe OSA, which the authors posit may lead to AD onset. 

The paper is well written and the statistics provided seem appropriately analysed. 

Author Response

We are grateful for your generous comments.

Thank you for your time and consideration in reviewing this manuscript.

Reviewer 2 Report

The authors present a case-control study about differential expressions of genes associated with Alzheimer Disease in ten middle-age patients with severe and non-severe Obstructive Sleep Apnea and without evident AD. Tissue samples for the study were taken from resected submucosal uvula after uvulopalatopharyngoplasty. After sequencing and gene expression and quantification analysis they found 4 EAOAD-associated genes and 56 LOAD-associated genes. As a result of the statistical analysis they found five AD-associated DEGs(differentially expressed genes) in patients with severe OSA that show enriched biological processes in the tissue: cell movement, secretion, enzyme production. This is the main contribution of the study. In addition, they found five significantly different expressions in patients with severe OSA compared with patients with non-severe OSA.

The paper title would benefit if the order/sense of the words change to be clear that the study is about the alterations of the gene expression’ associated with AD, otherwise it seems confuse to me as it reads that the patients are associated or have the Alzheimer’s Disease. Consider to add the study design into it.

The report explains successfully the aim of the study and the background for the investigation. Setting, locations and relevant dates including periods of recruitment and data collection are described as well as the source of case ascertainment and control selection (matching criteria). Nevertheless, it would be a recommendation to add references that support the validity of the use of the apnea-hypopnea index (AHI) to determine sever or non-severe OSA.

Statistical methods are well described, key results are well summarized and the discussion of the limitations of the study included.

The paper would benefit with a more detailed discussion about the generalizability of the study results and the implications for treatment of the OSA patients.

Line 201. Although the difference in the incidence of AD between …severe OSA. This paragraph is confuse to me.

Author Response

Response:

Thank you for your time and consideration in reviewing this manuscript. We totally agree and thank you for your valuable comments. We have revised the manuscript according to your comments.

Point 1:

The paper title would benefit if the order/sense of the words change to be clear that the study is about the alterations of the gene expression’ associated with AD, otherwise it seems confuse to me as it reads that the patients are associated or have the Alzheimer’s Disease.

Response to point 1:

Thank you for your valuable comments. We have retitled this paper to be “Alterations of Alzheimer's Disease-Associated Gene Expression in Severe Obstructive Sleep Apnea Patients”.

Point 2: Nevertheless, it would be a recommendation to add references that support the validity of the use of the apnea-hypopnea index (AHI) to determine severe or non-severe OSA.

Response to point 2:

Yes, the related reference was added (line 74–77, reference 43).

Point 3:

The paper would benefit with a more detailed discussion about the generalizability of the study results and the implications for treatment of the OSA patients. 

Response to point 3:

Yes. We have revised the manuscript based on your valuable suggestions (line 221–226).

Point 4:

Line 201. Although the difference in the incidence of AD between …severe OSA. This paragraph is confusing to me. 

Response to point 4:

Thank you for your excellent comment. We have rephrased the paragraph for more clarity (line 207–208).

Thank you again for your highly professional comments.

Reviewer 3 Report

Comments to the Authors

This study examined differential expressions of genes associated with AD in OSA patients without evident AD by prospectively recruiting five severe OSA patients that were age and sex-matched with five non-severe OSA without evident dementia who underwent uvulopalatopharyngoplasty between January 1, 2013 and December 31, 2015. RNA isolation was done. Gene quantification was performed using RNA-Seq by Expectation-Maximization and differentially expressed genes [DEGs] were screened using the Poisson distribution method. 63 genes including 4 Early Onset Alzheimer’s disease and 56 Late Onset Alzheimer’s disease associated genes were analyzed. Results from the study show the expressions of CCL2, IL6, CXCL8, HLA-A, and IL1RN in the patients with severe OSA significantly different from those in the patients with non-severe OSA and contributed to changes in the immune response, cytokine-cytokine receptor interactions, and nucleotide-binding oligomerization domain-like receptor signaling pathways. Authors conclude that inflammation may contribute to the onset of AD, and that physicians should be aware of the potential occurrence of AD in patients with severe OSA.

I think the paper has a well-justified underlying rationale; however, and was easy to read. The various sections of the paper from the introduction, methods, statistical analysis, results and discussions were well articulated. In general, the results and conclusion appear quite straightforward. I address some of the issues below:

Abstract

Minor Issue

The authors should state their objectives explicitly in the Objective/Background section. If the issue is with word count, they can do away with what is written already or rephrase to include the fact that prevalence data on patients with severe dementia is non-existent, and then explicitly state their objective. This should be clear from the beginning.

No information is given about the statistical analysis conducted. The reader would like to know up front if appropriate statistical analytic tools/methods was used.

Introduction

The introduction is concise and to the point, however authors need to improve on their citations regarding the association between OSA and AD. I have provided relevant citations that authors need to review and possibly incorporate. The evidence does not only show an association of OSA and AD risk but also associations of OSA with validated biomarkers of AD, including the presence of significant brain Aβ load, measured either by cerebrospinal (CSF) Aβ42 or PET amyloid imaging, and CSF levels of tau (i.e. total or phosphorylated), in both cognitive normal and mild cognitive impaired (MCI) participants, have been shown at cross-section and longitudinally. (See citations below)

Chang WP, Liu ME, Chang WC, et al. Sleep apnea and the risk of dementia: a population-based 5-year follow-up study in Taiwan. PLoS One. 2013;8(10):e78655. Osorio RS, Gumb T, Pirraglia E, et al. Sleep-disordered breathing advances cognitive decline in the elderly. Neurology. 2015;84(19):1964-1971. Yaffe K, Laffan AM, Harrison SL, et al. Sleep-disordered breathing, hypoxia, and risk of mild cognitive impairment and dementia in older women. JAMA : the journal of the American Medical Association. 2011;306(6):613-619. Yaffe K, Nettiksimmons J, Yesavage J, Byers A. Sleep Quality and Risk of Dementia Among Older Male Veterans. The American journal of geriatric psychiatry : official journal of the American Association for Geriatric Psychiatry. 2015;23(6):651-654. Spira AP, Yager C, Brandt J, et al. Objectively Measured Sleep and beta-amyloid Burden in Older Adults: A Pilot Study. SAGE open medicine. 2014;2. Yun CH, Lee HY, Lee SK, et al. Amyloid Burden in Obstructive Sleep Apnea. Journal of Alzheimer's disease : JAD. 2017. Ju YE, Finn MB, Sutphen CL, et al. Obstructive sleep apnea decreases central nervous system-derived proteins in the cerebrospinal fluid. Annals of neurology. 2016;80(1):154-159. Osorio RS, Ayappa I, Mantua J, et al. The interaction between sleep-disordered breathing and apolipoprotein E genotype on cerebrospinal fluid biomarkers for Alzheimer's disease in cognitively normal elderly individuals. Neurobiol Aging. 2014;35(6):1318-1324. Liguori C, Mercuri NB, Izzi F, et al. Obstructive Sleep Apnea is Associated With Early but Possibly Modifiable Alzheimer's Disease Biomarkers Changes. Sleep. 2017;40(5). Sharma RA, Varga AW, Bubu OM, et al. Obstructive Sleep Apnea Severity Affects Amyloid Burden in Cognitively Normal Elderly. A Longitudinal Study. American journal of respiratory and critical care medicine. 2018;197(7):933-943. Bubu OM, Pirraglia E, Andrade AG, et al. Obstructive Sleep Apnea and Longitudinal Alzheimer's disease biomarker changes. Sleep. 2019.

The risk of AD in patients with OSA based on a nationwide population-based cohort especially in Asia exist. (See below)

Chang WP, Liu ME, Chang WC, Yang AC, Ku YC, Pai JT, et al. Sleep apnea and the risk of dementia: a population-based 5-year follow-up study in taiwan. PLoS One. 2013;8(10):e78655. Lee JE, Yang SW, Ju YJ, Ki SK, Chun KH. Sleep-disordered breathing and Alzheimer's disease: A nationwide cohort study. Psychiatry research. 2019;273:624-30. Yun CH, Lee HY, Lee SK, Kim H, Seo HS, Bang SA, et al. Amyloid Burden in Obstructive Sleep Apnea. Journal of Alzheimer's disease : JAD. 2017.

 Moreover, meta-analysis data provide an aggregate of findings from multiple studies and have sub-group analyses examining OSA and AD or Dementia. (See below)

Bubu OM, Brannick M, Mortimer J, et al. Sleep, Cognitive impairment, and Alzheimer's disease: A Systematic Review and Meta-Analysis. Sleep. 2017;40(1). Shi L, Chen SJ, Ma MY, et al. Sleep disturbances increase the risk of dementia: A systematic review and meta-analysis. Sleep medicine reviews. 2018;40:4-16.

Methods and Results

Minor issue

Subjects

It is unclear where the participants were recruited from. Where they recruited from the clinic or were they post-surgical patients recruited via patient chart records?

Line 132 says the patient characteristics are shown in Table 1 in the supplement. However eTable 1. Shows Genes and pseudogenes associated with Alzheimer's disease.

Line 147 – 148: This observation implied that severe OSA elicited modest perturbations in the AD-associated transcriptional response of the oropharynx that did not vary from that elicited from non-severe OSA patients?

Discussion

Minor issue

Lines 98 – 199 please provide citation (see citations above)

Major issue

Please describe your limitations in a more elaborate manner and what you did if anything to mitigate the situation. What you have written does not necessarily articulate properly any limitation.

Conclusion

 Looks good

Tables and Figures: Looks good

Author Response

Response:

Thank you for your time and consideration in reviewing this manuscript. We have reviewed the English language and spell in the manuscript and have confirmed that the text words presented in the manuscript are without error. We totally agree and thank you for your valuable comments. We have revised the manuscript according to your comments.

Point 1:

[Abstract - Minor Issue] The authors should state their objectives explicitly in the Objective/Background section. If the issue is with word count, they can do away with what is written already or rephrase to include the fact that prevalence data on patients with severe dementia is non-existent, and then explicitly state their objective. This should be clear from the beginning.

No information is given about the statistical analysis conducted. The reader would like to know up front if appropriate statistical analytic tools/methods was used. 

Response to point 1:

Many thanks for the excellent comments. We have revised the abstract based on your comments (line 31–40).

Point 2:

[Introduction] The introduction is concise and to the point, however authors need to improve on their citations regarding the association between OSA and AD. I have provided relevant citations that authors need to review and possibly incorporate. The evidence does not only show an association of OSA and AD risk but also associations of OSA with validated biomarkers of AD, including the presence of significant brain Aβ load, measured either by cerebrospinal (CSF) Aβ42 or PET amyloid imaging, and CSF levels of tau (i.e. total or phosphorylated), in both cognitive normal and mild cognitive impaired (MCI) participants, have been shown at cross-section and longitudinally.

Response to point 2:

Thank you for your valuable comments. We have revised the introduction section and added related references (line 61–64).

Point 3:

The risk of AD in patients with OSA based on a nationwide population-based cohort especially in Asia exist.

Response to point 3:

Thank you for addressing this important concern. We have revised this paragraph and added the reference (line 51–52).

Point 4:

Moreover, meta-analysis data provide an aggregate of findings from multiple studies and have sub-group analyses examining OSA and AD or Dementia.

Response to point 4:

Yes. We have revised this paragraph and added citations (line 51–52).

Point 5:

[Methods and Results - Minor issue – Subjects] It is unclear where the participants were recruited from. Where they recruited from the clinic or were they post-surgical patients recruited via patient chart records? 

Response to point 5:

Yes. We have revised this paragraph based on your comments (line 74–77).

Point 6:

Line 132 says the patient characteristics are shown in Table 1 in the supplement. However eTable 1. Shows Genes and pseudogenes associated with Alzheimer's disease.

Response to point 6:

Thank you for your valuable comment. We have corrected this sentence (line 140).

Point 7:

Line 147 – 148: This observation implied that severe OSA elicited modest perturbations in the AD-associated transcriptional response of the oropharynx that did not vary from that elicited from non-severe OSA patients?

Response to point 7:

Yes, thanks for your cogent comment. We have revised the paragraph for more clarity (line 153–156).

Point 8:

[Discussion - Minor issue] Lines 198 – 199 please provide citation. 

Response to point 8:

Thank you for the great comment. The related citations are provided (line 204–205).

Point 9:

[Major issue] Please describe your limitations in a more elaborate manner and what you did if anything to mitigate the situation. What you have written does not necessarily articulate properly any limitation.

Response to point 9:

Thank you for addressing this important concern. We have rephrased the discussion section and described the limitations (line 217–226).

Thank you again for your highly professional comments.